# Cloud Computing Research Profiling: Mapping Scholarly Community and Identifying Thematic Boundaries of the Field

**Agata Sudolska *** , **Andrzej Lis *** and **Róża Błaś**

Faculty of Economic Sciences and Management, Nicolaus Copernicus University in Toruń, 87-100 Toruń, Poland; roza.blas@gmail.com

* Correspondence: aga@econ.umk.pl (A.S.); andrzejlis@econ.umk.pl (A.L.)

**Abstract:** The aim of the study was to map the scholarly community interested in research on cloud computing and to identify thematic boundaries of the field. The methodology of research profiling, representing bibliometric descriptive studies, was applied to achieve the aim of the study. Using research profiling for mapping the cloud computing field can be considered as an innovation. Although the research profiling methodology has been widely used across various subject areas, including Computer Science, Social Sciences, Engineering, Arts and Humanities, Business, Management and Accounting, and Psychology, thus far neither Scopus nor Web of Science indexed publications including the conjunction of phrases "cloud computing" and "research profiling" in their titles, keywords and abstracts. The previous important scientometric study of the research output in the field was published by Heilig and Voß in 2014. Taking into account a very dynamic growth of the field, all this indicates the research gap to be filled. The research sample is made of 14,158 publications indexed in Scopus database comprising the phrase "cloud computing" in their titles. The study was purposely limited to the title search to concentrate the attention of publications relating directly to the issue of cloud computing. Applying the quantitative approach provides an opportunity for broad scanning of subject-related literature. First, general publication profiling recognized the main contributors (countries, research intuitions, source titles and authors) to the scholarly community interested in cloud computing. Secondly, subject area profiling was applied to find how multidisciplinary is the research in the field and how the research output is distributed across subject areas. Finally, topic profiling unveiled leading topics of studies in the field and their distribution by authors, journal, subject areas and core references.

**Keywords:** cloud computing; bibliometrics; research profiling

## 1. Introduction

New technologies attract researchers' attention steaming not only from their economic, political and socio-cultural influence but also, among other things, from their self-generating process, clearly evident in this day and age. One of the outcomes of such a process—where one technological solution or a combination of several of them gives birth to a new one—is cloud computing. As a set of various technologies, being developed since the 1960s, cloud computing has rapidly transformed the way of viewing computing resources in the 21st century (Buyya et al. 2008). Utilizing homogenous and global networks based on common assets and protocols (Subashini and Kavitha 2011), cloud computing has flourished as a next utility power (Buyya et al. 2009). By making the dawn of digitizing businesses, government agencies, financial institutions, schools and universities, medical facilities, households, and other market players (Foster et al. 2008; Olczak 2014; Mostafa et al. 2019), it elevated to a rank

of a global phenomenon. Information technology resources such as hardware and software never were as available as now (Zhang et al. 2010; Kopishynska et al. 2016). This opened intense debate among researchers and practitioners surrounding a wide variety of areas relating to cloud computing, possibilities it creates, and influence on human life it makes.

Despite the rapidly growing popularity of studies on cloud computing, the related research field seems to be insufficiently mapped, at least as regards the research field profiling. Porter et al. (2002) recommended using the method of research profiling to scan the body of the literature in search of research fronts and thematic boundaries. Research profiling method is aimed at discovering "What issues are central? What techniques are emphasised? Who constitutes the scholarly community engaged in this particular research domain? When—how is the research domain evolving over time" (Porter et al. 2002). We tested whether the very abundant research in cloud computing has been ever analyzed with the method of research profiling. As of 20 July 2018, neither Scopus nor Web of Science indexed publications including the conjunction of phrases "cloud computing" and "research profiling" in titles, keywords and abstracts. However, we are aware of an extensive scientometric study in context of cloud computing by Heilig and Voß (2014). The authors investigated over 15,000 publications on cloud computing published in the period 2008–2013. Taking into account that cloud computing is the area that develops rapidly nowadays, we found it necessary to update the scientometric study in this field and find out eventual changes.

Having identified the aforementioned gap in the research field, we decided to fill it through profiling of the research output related to the issue of cloud computing. The aim of the study was to map the scholarly community interested in research on cloud computing and to identify thematic boundaries of the field. The following research questions were asked to operationalize the aim of the study: (1) What are the main contributors to the scholarly community interested in cloud computing? (2) How much multidisciplinary is research in the field and how is the research output distributed across subject areas? (3) What are the leading topics of studies in the field?

## 2. Method of Study

### 2.1. Research Methodology

Research profiling is categorized among descriptive bibliometric studies and recommended to support classical literature surveys. As summarized by (Porter et al. 2002), "[t]his broad scan of contextual literature can extend the span of science by better linking efforts across research domains. Topical relationships, research trends, and complementary capabilities can be discovered". The overview of publications indexed in Scopus and Web of Science shows a wide use of research profiling methodology across various subject areas including: Computer Science, Social Sciences, Engineering, Arts and Humanities, Business, Management and Accounting, and Psychology. The method was among others used to analyze and profile research in such fields as: nano-enhanced, thin-film solar cells (Guo et al. 2010), standardization and innovation (Choi et al. 2011), multiple criteria decision making (Bragge et al. 2012), relative absorptive capacity (Martinez et al. 2012), or the learning organization (Lis 2017).

The research profiling methods contain the following components (Martinez et al. 2012; Lis 2018; cf. Choi et al. 2011):

- general publication (research) profiling aimed at mapping the scholarly community in the field, including leading contributing countries, research institutions, source titles (journals and other types of publications) and authors;
- subject area profiling exploring multidisciplinary character of the field and covering the analysis of distribution of leading source titles, authors and core references across the variety of subject areas in the field; and
- topic profiling oriented to discovering leading topics typical of the key contributing source titles, authors, subject areas and core references.

We applied the aforementioned model while designing the research procedure of our study and the structure of the paper.

## 2.2. Research Sample

While sampling for the study, the phrase "cloud computing" was searched in titles of publications indexed in Scopus database. The query was purposely limited to the title search to concentrate the attention on publications relating directly to the issue of cloud computing. For the same reason, we also excluded any other expressions linked with the topic, e.g., "hybrid cloud" or "cloud federation". Nevertheless, we are aware of limitations resulting from such a design of the research sampling process, which are explained in details in Section 6.

As of 20 July 2018, 14,158 publications were retrieved, which had received in total 147,741 citations. The h-index for the sample is 151. Scientific productivity on cloud computing and its yearly distribution are presented in Figure 1.

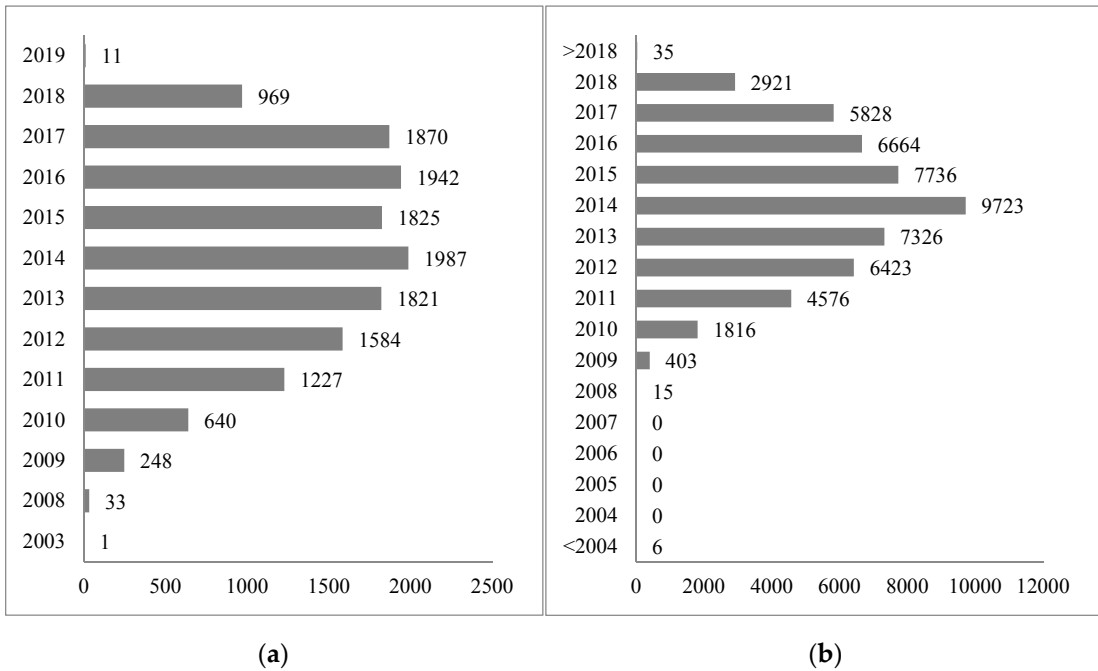

**(a)**　　　　　　　　　　　　　　　　　　　　　　**(b)**

**Figure 1.** Scientific productivity of research on cloud computing: (**a**) number of publications; and (**b**) number of citations. Source: Own study based on data retrieved from Scopus database (accessed 20 July 2018).

As the data in Figure 1 show, cloud computing is a very hot topic in contemporary research. The first work included in the sample was published 15 years ago (Barry 2003). However, in fact, the emergence of cloud computing studies dates back to 2008/2009. Analyzing the number of publications issued between 2008 and 2017, two periods may be distinguished: (1) breakthrough and fast growth (from 33 to 1584 papers) between 2008 and 2012; and (2) stabilized high research interest in the topic (with the output ranging from 1821 to 1987 items) between 2013 and 2017. As regards citations of the research output, very similar trends may be observed. The clear peak in the number of received citations in 2014 is a very distinctive feature of the research sample.

## 3. General Publication Profiling: Mapping the Scholarly Community

### 3.1. Country Profiling

The research productivity in cloud computing is distributed among 127 countries/territories. In total, 500 records in the research sample are of undefined country/territory. Among the leaders

(Top 10 most productive countries/territories), there are nations representing Asia, Australia, Europe and North America (cf. Table 1). The frontrunners in South America and Africa are, respectively, Brazil (N = 223, Rank 15) and Morocco (N = 161, Rank 18).

**Table 1.** Most productive countries/territories in research on cloud computing.

| No. | Country/Territory | Publications | | Citations | | |
|---|---|---|---|---|---|---|
| | | N | % | N | % | h-index |
| 1. | China | 4182 | 29.5 | 26,810 | 18.1 | 68 |
| 2. | India | 2210 | 15.6 | 9589 | 6.5 | 37 |
| 3. | United States | 1887 | 13.3 | 46,143 | 31.2 | 92 |
| 4. | United Kingdom | 619 | 4.4 | 9170 | 6.2 | 45 |
| 5. | South Korea | 572 | 4.0 | 5238 | 3.5 | 34 |
| 6. | Taiwan | 482 | 3.4 | 3812 | 2.6 | 27 |
| 7. | Australia | 462 | 3.3 | 19,513 | 13.2 | 53 |
| 8. | Germany | 418 | 3.0 | 5923 | 4.0 | 37 |
| 9. | Canada | 364 | 2.6 | 7183 | 4.9 | 37 |
| 10. | Malaysia | 356 | 2.5 | 4541 | 3.1 | 32 |

Own study based on data retrieved from Scopus database (accessed 20 July 2018).

Among the most productive countries in regard to both quantity and visibility of publications, are China, India and the United States. However, although China and India are more productive in terms of quantity, the United States is the leader in terms of their visibility. Overall, 4182 publications (constituting 29.5% of the research sample) affiliated by Chinese institutions received 18.1% of all citations (N = 26,810). While considering productivity of India, 2210 publications (constituting 15.6% of the research sample) affiliated by Indian institutions were cited 9589 times (6.5% of all citations, N = 26,810). Thus, in total, Chinese and Indian publications make 24.6% of all citations in the field. In both cases, high values of h-index are observed: 68 for China and 37 for India. As far as the publications affiliated by American institutions are concerned, it is worth noting that 1887 works (13.3% of the research sample) received 31.2% of all citations. Publications affiliated by American institutions were cited 46,143 times. Moreover, one can notice that, due to a high value of h-index (92), the United States is a leader in regard to the visibility of the work. The remaining most productive countries can be grouped in three categories. The first category includes the United Kingdom and South Korea. These countries contribute to the field of study with more than 500 publications per country (619 and 572, respectively). What is noticeable, the United Kingdom overtakes South Korea in terms of publication visibility. The publications affiliated by British institutions received almost twice more citations than the publications affiliated by South Korean institutions (9170 and 5238 citations, respectively). In addition, in the case of publications affiliated by British institutions, one can observe a higher value of h-index (45 in comparison to 34 for South Korea). The second category includes, e.g., Taiwan, Germany, Canada and Malaysia, with the output below 500 items per country. In this group, the number of citations assigned to a particular country varies between 7183 (the highest number for Canada) and 3812 (the lowest number for Taiwan). In the case of h-index, the above mentioned countries are characterized by values from 27 (the lowest result for Taiwan) to 37 (the highest result for Germany and Canada). The third category is constituted separately by Australia. In terms of quantity, its research output represents the same level as publications of aforesaid countries of the second category. However, while taking visibility perspective, one can notice that publications affiliated by Australian institutions received 19,513 citations with a high value of h-index (53).

### 3.2. Institution Profiling

The Top 10 most productive research institutions contributed to the field with 1171 publications (1237 while calculated separately one by one), which makes 8.27% (respectively, 8.74%) of the research sample. This output received in total 22,959 citations (23,831 calculated separately), i.e., 15.54%

(respectively, 16.13%). The h-index for this group is 58. The most productive institutions in the field are ranked in Table 2. As the catalogue is dominated by Chinese organizations, we added to the table information about the top leading universities from the most productive countries listed in Table 1, regardless of the position these institutions occupy.

**Table 2.** Top most productive institutions in research on cloud computing.

| No. | Institution | Country/ Territory | Publications | | Citations | | |
|---|---|---|---|---|---|---|---|
| | | | N | % | N | % | h-Index |
| 1. | Beijing University of Posts and Telecommunications | China | 195 | 1.38 | 997 | 0.67 | 16 |
| 2. | Chinese Academy of Sciences | China | 158 | 1.12 | 1762 | 1.19 | 18 |
| 3. | Vellore Institute of Technology | India | 131 | 0.93 | 553 | 0.37 | 12 |
| 4. | Tsinghua University | China | 126 | 0.89 | 2289 | 1.55 | 22 |
| 5. | Xidian University | China | 121 | 0.85 | 1083 | 0.73 | 17 |
| 6. | Ministry of Education of China | China | 116 | 0.82 | 1129 | 0.76 | 16 |
| 7–8. | Huazhong University of Science and Technology | China | 102 | 0.72 | 987 | 0.67 | 16 |
| 7–8. | Wuhan University | China | 102 | 0.72 | 802 | 0.54 | 16 |
| 9. | University of Melbourne | Australia | 96 | 0.68 | 13,529 | 9.16 | 39 |
| 10. | Kyung Hee University | South Korea | 90 | 0.64 | 700 | 0.47 | 15 |
| 16. | University of Malaya | Malaysia | 72 | 0.51 | 2617 | 1.77 | 26 |
| 36. | Arizona State University | United States | 48 | 0.34 | 1817 | 1.23 | 19 |
| 52. | National Cheng Kung University | Taiwan | 39 | 0.28 | 128 | 0.09 | 7 |
| 58. | University of Waterloo | Canada | 35 | 0.25 | 2208 | 1.49 | 16 |
| 114–122 | University of Southampton | United Kingdom | 25 | 0.18 | 345 | 0.23 | 10 |
| 114–122 | Leeds Becket University | United Kingdom | 25 | 0.18 | 339 | 0.23 | 8 |
| 114–122 | Technical University of Berlin | Germany | 25 | 0.18 | 244 | 0.17 | 8 |

Own study based on data retrieved from Scopus database (accessed 20 July 2018).

All the most productive research institutions represent the countries identified as the main contributors to the research field. Among them, the leaders in terms of quantity and quality of research are: Beijing University of Posts and Telecommunications and Chinese Academy of Sciences. The output of Beijing University of Posts and Telecommunications' researchers is 195 publications that gathered 997 citations and the value of h-index equals 16. The researchers of Chinese Academy of Sciences produced 158 works cited in 1762 publications (h-index value equal 18). In addition, Tsinghua University (China) needs to be highlighted due to a high number of received citations (N = 2289) as well as the h-index value (22).

It is interesting that nine of the ten most productive institutions are located in Asia (in China or South Korea). Only one institution among ten top ones (the University of Melbourne) is located in Australia. The University of Melbourne is noticeable as its output in the field, although includes only 96 works, has received 13,529 citations and its h-index value equals 39. Given that the United States is the leader in terms of visibility of the publications in the field, it is surprising that the most productive American university (i.e., Arizona State University) is ranked 36th.

We assume that the dominant position occupied by China (Table 1) and Chinese research institutions results from the fact that, in the last two decades, the Chinese government has prioritized the development of cloud computing technology as well as the research in this field and therefore allocated significant resources to this area (Ragland et al. 2013). While considering India's great activity

in the field of cloud computing research (Table 1), it might be related to a very quick development of cloud computing services and technologies in India as well as it might result from rapid growth of employment in this field observed both in China and India (Wadhe 2016).

South Korea is one of the most stable markets for cloud computing due to existing infrastructure and government financing of ICT and cloud expansion. In South Korea, the rate of IT spending has been growing at a rate of 4.5 times since 2009, and it is positively anticipated to end up growing at a rate of better than six times more until 2020, boosting total spending on cloud computing from $67 billion in 2015 to $162 billion in 2020 (Seok-Keun and Bo-Young 2018). Moreover, Asian countries research in the field of cloud computing is supported by Asia Cloud Computing Association (www. asiacloudcomputing.org).

*3.3. Source Title Profiling*

The Top 10 most productive source titles contributed to the field with 2135 publications (i.e., 15.08% of the research sample), which received in total 14,712 citations (9.96%). Their h-index value is 44. The most productive source titles in the field are ranked in Table 3.

**Table 3.** Top most productive source titles in research on cloud computing.

| No. | Source Title | Publications | | Citations | | |
|---|---|---|---|---|---|---|
| | | N | % | N | % | h-Index |
| 1. | Lecture Notes in Computer Science | 461 | 3.26 | 4060 | 2.75 | 27 |
| 2. | Applied Mechanics and Materials | 296 | 2.09 | 149 | 0.10 | 5 |
| 3. | Communications in Computer and Information Science | 243 | 1.72 | 755 | 0.51 | 12 |
| 4. | Advances in Intelligent Systems and Computing | 238 | 1.68 | 380 | 0.26 | 9 |
| 5. | Advanced Materials Research | 187 | 1.32 | 93 | 0.06 | 4 |
| 6. | ACM International Conference Proceedings Series | 179 | 1.26 | 734 | 0.50 | 12 |
| 7. | Lecture Notes in Electrical Engineering | 172 | 1.21 | 297 | 0.20 | 7 |
| 8. | International Journal of Applied Engineering Research | 125 | 0.88 | 61 | 0.04 | 4 |
| 9. | Future Generation Computer Systems | 119 | 0.84 | 8183 | 5.54 | 28 |
| 10. | Procedia Computer Science | 115 | 0.81 | 811 | 0.55 | 16 |

Own study based on data retrieved from Scopus database (accessed 20 July 2018).

The leading prominent source title in research on cloud computing is *Lecture Notes in Computer Science*. In total, 461 works in the field have been published in this source, receiving 4060 citations with the h-index value of 27. In regard to the quantity of publications, *Applied Mechanics and Materials*, *Communications in Computer and Information Science* and *Advances in Intelligent Systems and Computing* should be highlighted. In each source title, there have been over 200 works published on cloud computing (296, 243 and 238, respectively). H-index value for these sources varies from 5 to 12 (5, 12 and 9, respectively). In all remaining most productive journals, fewer than 200 works in the field of cloud computing have been published. What is of particular interest, the number of citations in this group of source titles varies from 61 to 8183 and h-index from 4 to 28. Among these source titles, *Future Generation Computer Systems* must be highlighted as outstanding due to the fact that just 119 work published in this source title were cited 8183 times. In addition, their h-index value is very high, equal to 28. Thus, in regard to visibility of publications in the field *Future Generation Computer Systems* is much above average in this group.

*3.4. Author Profiling*

The most prolific authors contributed to the field with 355 publications (382 while calculated separately one by one), which makes 2.51% (respectively, 2.70%) of the research sample. Their output received in total 18,900 citations (20,359 calculated separately) i.e., 12.79% (respectively, 13.78%). The h-index for the collection of top authors' publications is 54.

Among the most prolific researchers dealing with the issue of cloud computing, the most attention is received by the works of R. Buyya from the University of Melbourne, which were cited 12,962 times (i.e., 8.77% of all citations in the research sample) and their h-value is 38. His most influential works comprise publications focused on distinguishing cloud computing from other technologies, applying cloud computing in business as well as pointing out challenges of using cloud computing and optimizing cloud architecture. R. Buyya is followed by E.N. Huh from Kyung Hee University (48 publications) and A. Gani representing the University of Malaya (41 publications). While analyzing the productivity of these two authors, one can notice that, although E.N. Huh has produced 48 works in the field of cloud computing, – he gathered only 343 citations and his h-index value is 8. On the other hand, A. Gani's publications were cited 1952 times and -his h-index value is much higher, equal to 19. Taking into account the number of publications and citations, J. Li from Guangzhou University as well as A. V. Vasilakos from Lulea University of Technology in Sweden should be mentioned. J. Li's output in the field encompasses 34 publications that received 1158 citations and their h-index value is 15. A.V. Vasilakos has produced fewer works (29) but they were cited more often (1621 times) and their h-index value equals 16. The remaining most prolific authors presented in Table 4 contributed to the field with 27 or fewer works, gathering from 126 to 502 citations. The h-index value in this group varies from 7 to 12.

**Table 4.** Top most productive authors in research on cloud computing.

| No. | Author | Institution | Country | Publications | | Citations | | |
|---|---|---|---|---|---|---|---|---|
| | | | | N | % | N | % | h-Index |
| 1. | Buyya, R. | University of Melbourne | Australia | 80 | 0.57 | 12,962 | 8.77 | 38 |
| 2. | Huh, E.N. | Kyung Hee University | South Korea | 48 | 0.34 | 343 | 0.23 | 8 |
| 3. | Gani, A. | University of Malaya | Malaysia | 41 | 0.29 | 1952 | 1.32 | 19 |
| 4. | Li, J. | Guangzhou University | China | 34 | 0.24 | 1158 | 0.78 | 15 |
| 5. | Vasilakos, A.V. | Lulea University of Technology | Sweden | 29 | 0.20 | 1621 | 1.10 | 16 |
| 6. | Qiu, M. | Shenzen University | China | 27 | 0.19 | 456 | 0.31 | 11 |
| 7–9. | Chen, X. | Xidian University | China | 25 | 01.8 | 502 | 0.34 | 12 |
| 7–9. | Jin, H. | Huazhong University of Science and Technology | China | 25 | 0.18 | 398 | 0.27 | 9 |
| 7–9. | Rong, C. | Stavanger University | Norway | 25 | 0.18 | 386 | 0.26 | 7 |
| 10–11. | Chang, V. | Xi'an Jiaotong–Liverpool University | China | 24 | 0.17 | 455 | 0.31 | 10 |
| 10–11. | Park, J.H. | Seoul National University of Science and Technology | South Korea | 24 | 0.17 | 126 | 0.09 | 7 |

Own study based on data retrieved from Scopus database (accessed 20 July 2018).

General publication profiling has identified the most prominent contributors to the research field focused on cloud computing, including the most productive countries, leading research institutions and source titles as well as the most prolific authors. Subject area profiling was the next step of the study procedure. Its aim was to explore multidisciplinary character of research in the field. The idea was to find how the leading journals, the most prolific authors and core references are distributed among subject areas. Moreover, subject area profiling was to show similarities and differences between various research perspectives in the field associated with a given subject area.

## 4. Subject Area Profiling: Exploring Multidisciplinary Character of Research in the Field

The publications comprising the research sample are distributed among 25 subject areas. The six leading categories, selected as the subject of a thorough subject area profiling, are: Computer Science (10,627), Engineering (4488), Mathematics (2173), Social Sciences (884), Business, Management

and Accounting (665), and Decision Sciences (655). The remaining subject areas are: Physics and Astronomy (345), Medicine (328), Materials Science (257), Earth and Planetary Sciences (207), Energy (205), Biochemistry, Genetics and Molecular Biology (200), Environmental Science (195), Economics, Econometrics and Finance (146), Multidisciplinary (145), Agricultural and Biological Sciences (98), Chemistry (82), Pharmacology, Toxicology and Pharmaceutics (76), Chemical Engineering (72), Health Professions (60), Arts and Humanities (48), Psychology (22), Immunology and Microbiology (16), Neuroscience (12), Nursing (9), and Veterinary (1). Subject area profiling of research on cloud computing focused on identifying top contributing source titles, most prolific authors, and core references in six leading subject areas in the field i.e., Computer Science, Engineering, Mathematics, Social Sciences, Business, Management and Accounting, and Decision Sciences.

### 4.1. Source Title—Subject Area Profiling

To verify whether key subject areas are associated to some specific source titles, top most productive journals in each of key subject areas are listed in Table 5.

**Table 5.** Top most productive source titles in research on cloud computing by a subject area.

| Subject Area | Journals |
|---|---|
| Computer Science (10,627) | Lecture Notes in Computer Science (461), Communications in Computer and Information Science (243), Advances in Intelligent Systems and Computing (238), ACM International Conference Proceedings Series (179), Future Generation Computer Systems (119), Procedia Computer Science (115), Journal of Supercomputing (99), Cluster Computing (97), Journal of Theoretical and Applied Information Technology (70), Lecture Notes of the Institute for Computer Sciences, Social Informatics and Telecommunications Engineering (70) |
| Engineering (4488) | Applied Mechanics and Materials (296), Advances in Intelligent Systems and Computing (238), Advanced Materials Research (187), Lecture Notes in Electrical Engineering (172), International Journal of Applied Engineering Research (125), IEEE Access (45), International Journal of Pharmacy and Technology (37), Proceedings IEEE INFOCOM (37), Proceedings of the Annual Hawaii International Conference on System Sciences (34), Journal of Advanced Research in Dynamical and Control Systems (31) |
| Mathematics (2173) | Lecture Notes in Computer Science (461), Communications in Computer and Information Science (117), Journal of Supercomputing (99), International Journal of Pure and Applied Mathematics (93), Journal of Theoretical and Applied Information Technology (70), Concurrency Computing (46), Proceedings of the International Conference on Cloud Computing Technology and Science (43), Mathematical Problems in Engineering (29), Advanced Science Letters (27), Journal of Computational and Theoretical Nanoscience (27), Proceedings of SPIE The International Society for Optical Engineering (27) |
| Social Sciences (884) | Advanced Science Letters (27), Proceedings of the 10th INDIACom: 2016 3rd International Conference on Computing for Sustainable Global Development (23), International Journal of Information Management (20), Cloud Computing and Government Background Benefits Risks (17), International Journal of Emerging Technologies in Learning (15) |
| Business, Management and Accounting (665) | Wit Transactions on Information and Communication Technologies (30), Lecture Notes in Business Information Processing (24), International Journal of Business Information Systems (15), International Journal of Grid and Utility Computing (14), Computer Law and Security Review (9), Enterprise Management Strategies in the Era of Cloud Computing (9), Journal of Network and Systems Management (9) |
| Decision Sciences (655) | IEEE Transactions on Services Computing (29), Information Sciences (24), Lecture Notes in Business Information Processing (24), IFIP Advances in Information and Communication Technology (19), International Journal of Business Information Systems (15) |

Own study based on data retrieved from Scopus database (accessed 20 July 2018).

The analysis of the top most productive source titles associated with distinguished key subject areas allows stating that, for each subject area, there is a group of dedicated journal. Few of the most productive source titles are associated with more than one subject area (and no more than two of them). *Lecture Notes in Computer Science*, *Communications in Computer and Information Science*, *Supercomputing* and *Journal of Theoretical and Applied Information Technology* are the top-ranked journals in terms of the number of publications in two areas: Computer Science and Mathematics. However, *Lecture Notes in Computer Science* is the unquestionable leader as regards the number of published works (461). On the

other hand, *Advances in Intelligent Systems and Computing* is categorized as a leading journal in such subject areas under the study as Computer Science and Engineering. In addition, the areas of Business, Management and Accounting as well as Decision Science have the same most cited journals" *Lecture Notes in Business Information Processing* and *International Journal of Business Information Systems*. In the case of Social Sciences, one can notice that mostly it includes journals assigned only to it. There is only one source title—*Advanced Science Letters*—shared by Social Sciences area and Mathematics subject area. This allows assuming that the context of the publications on cloud computing assigned to Social Sciences differs from other main subject areas that have been distinguished.

### 4.2. Authors—Subject Area Profiling

Some subject areas and some authors are more productive than others. Table 6 depicts the most productive authors within the most prominent subject areas of research on cloud computing.

**Table 6.** Top most productive authors in research on cloud computing by a subject area.

| Subject Area | Authors |
|---|---|
| Computer Science (10,627) | Buyya, R. (73), Huh, E.N. (40), Li, J. (34), Gani, A. (31), Vasilakos, A.V. (27), Qiu, M. (26), Chen, X. (25), Rong, C. (24), Chang, V. (22), Jararweh, Y. (21) |
| Engineering (4488) | Yamada, S. (12), Buyya, R. (11), Liu, X. (11), Tamura, Y. (11), Vasilakos, A.V. (11), Gani, A. (10), Huh, E.N. (10), Pattnaik, P.K. (10), Yang, Z. (10), Yu, F.R. (9) |
| Mathematics (2173) | Buyya, R. (22), Gani, A. (13), Rong, C. (11), Chen, J. (10), Chen, X. (10), Huh, E.N. (9), Li, J. (9), Qiu, M. (9), Vasilakos, A.V. (9), Jin, H. (8), Park, J.H. (8), Zhan, Z.H. (8) |
| Social Sciences (884) | Yuvaraj, M. (7), Sultan, N. (6), Chao, L. (4), Walters, R.J. (4) |
| Business, Management and Accounting (665) | Ratten, V. (10), Jha, M.K. (5), Priyadarshinee, P. (5), Raut, R.D. (5), Teuteberg, F. (5) |
| Decision Sciences (655) | Huh, E.N. (6), Fiorese, A. (5), Qiu, M. (5), Feuerlicht, G. (4), Gai, K. (4), Jha, M.K. (4), Kumar, P. (4), Priyadarshinee, P. (4), Raut, R.D. (4) |

Own study based on data retrieved from Scopus database (accessed 20 July 2018).

According to Table 6, the most productive author is R. Buyya (106). He has the largest number of publications considering both all listed subject areas together and such subject fields as Computer Science (73) and Mathematics (22). The next most prolific researchers are E.N. Huh (65) and A. Gani (54). E.N. Huh is the only author whose publications are assigned to four subject areas, namely Computer Science, Engineering, Mathematics, and Decision Sciences. He is also the most productive researcher in the case of Decision Sciences. After E.N. Huh, only R. Buyya, A. Gani, and A.V. Vasilakos are found among the most prolific researchers in three subject categories, and they are as follows: Computer Science, Engineering, and Mathematics. Apart from them, J. Li, M. Qiu, X. Chen, and C. Rong publish in the field of Computer Science and Mathematics on par with M. K. Jha, P. Priyadarshinee, and R.D. Raut in Business, Management and Accounting, and Decision Sciences. These last three researchers have the same number of publications in the two mentioned subject fields. Currently, however, V. Ratten (10) is the most prolific author in the area of Business, Management and Accounting. S. Yamada (12), in turn, has made the greatest contribution to Engineering, as did M. Yuvaraj (7) to Social Sciences.

### 4.3. Core References—Subject Area Profiling

To analyze distribution of core references across subject areas and interrelatedness among them, we selected the 10 most cited references from the areas of Computer Science (ComSci), and Engineering (Eng), and the five most cited publications categorized in Mathematics (Math), Social Sciences (SocSci),

Business Management and Accounting (BMA), and Decision Sciences (DecSci). Top ranking papers are marked with relevant numbers in Table 7. Moreover, we marked with "X" publications indexed in given subject areas but not included within the Top 10 or 5 works.

**Table 7.** Core references in research on cloud computing by a subject area.

| Reference | Citations [N] | Subject Area Rank | | | | | |
|---|---|---|---|---|---|---|---|
| | | ComSci | Eng | Math | SocSci | BMA | DecSci |
| Armbrust et al. (2010) | 4512 | 1 | - | - | - | - | - |
| Buyya et al. (2009) | 2963 | 2 | - | - | - | - | - |
| Calheiros et al. (2011) | 1816 | 3 | - | - | - | - | - |
| Foster et al. (2008) | 1627 | 4 | - | - | - | - | - |
| Zhang et al. (2010) | 1383 | 5 | - | - | - | - | - |
| Subashini and Kavitha (2011) | 1148 | 6 | - | - | - | - | - |
| Beloglazov et al. (2012) | 1109 | 7 | - | - | - | - | - |
| Buyya et al. (2008) | 1078 | 8 | 1 | - | - | - | - |
| Nurmi et al. (2009) | 1050 | 9 | - | - | - | - | - |
| Marston et al. (2011) | 937 | 10 | - | - | - | 1 | 1 |
| Yu et al. (2010) | 844 | X | 2 | - | - | - | - |
| Dinh et al. (2013) | 725 | X | 3 | - | - | - | - |
| Xu (2012) | 702 | X | 4 | 1 | - | - | - |
| Wang et al. (2010) | 657 | X | 5 | - | - | - | - |
| Takabi et al. (2010) | 596 | X | 6 | - | 1 | - | - |
| Buyya et al. (2010) | 495 | X | - | 2 | - | - | - |
| Wang et al. (2009) | 480 | X | - | 3 | - | - | - |
| Dillon et al. (2010) | 451 | - | 7 | - | - | - | - |
| Li et al. (2010) | 445 | X | 8 | - | - | - | - |
| Baliga et al. (2011) | 438 | - | 9 | - | - | - | - |
| Wang et al. (2009) | 428 | - | 10 | - | - | - | - |
| Sultan (2010) | 409 | X | - | - | 2 | - | - |
| Kaufman (2009) | 395 | X | X | - | 3 | - | - |
| Low et al. (2011) | 339 | X | X | - | - | 2 | - |
| Wang et al. (2010) | 319 | X | - | 4 | - | - | - |
| Grobauer et al. (2011) | 300 | X | X | - | 4 | - | - |
| Chaisiri et al. (2012) | 298 | X | - | - | - | - | 2 |
| Wei et al. (2010) | 265 | X | - | 5 | - | - | - |
| Wang et al. (2012) | 255 | X | - | - | - | - | 3 |
| Wei et al. (2014) | 239 | X | X | X | - | - | 4 |
| Ali et al. (2015) | 209 | X | X | X | - | - | 5 |
| Oliveira et al. (2014) | 176 | X | - | - | - | 3 | X |
| Gupta et al. (2013) | 175 | X | - | - | 5 | - | - |
| Alshamaila et al. (2013) | 151 | X | - | - | - | 4 | X |
| Venters and Whitley (2012) | 146 | X | - | - | X | 5 | - |

Own study based on data retrieved from Scopus database (accessed 20 July 2018).

Due to the nature of cloud computing, most of the research works listed in Table 7 cover a wide range of challenges related to cloud computing. These include Dillon et al. (2010), Buyya et al. (2010), Calheiros et al. (2011), and Baliga et al. (2011). Another example is Marston et al. (2011) who, among other things, listed weaknesses and threats of utilizing cloud computing. Solutions aimed at optimizing distributed computing resources, in turn, are a subject of such publications by Nurmi et al. (2009), Wei et al. (2010), Chaisiri et al. (2012), and Beloglazov et al. (2012). The challenges of using cloud computing also are deliberated in the most popular publication on cloud computing taking the first place in Computer Science. The article titled "A view of the cloud computing" by Armbrust et al. (2010) presents a wide range of obstacles and opportunities of deploying cloud computing in view of its peculiarities.

Closer inspection of Table 7 also shows that only three of the references are not present in Computer Sciences, namely those by Dillon et al. (2010), Baliga et al. (2011), and Wang et al. (2009). Moreover, more of the publications listed in Table 7, namely 24 positions, are shared between at least

two subject areas. For instance, the papers by Oliveira et al. (2014) and Alshamaila et al. (2013), aiming at identifying factors affecting cloud computing adoption, are shared by Computer Science, Business Management and Accounting, and Decision Sciences. This topic in a view of either businesses or individuals was also covered by Gupta et al. (2013), Low et al. (2011), and Venters and Whitley (2012).

Regardless the subject area, security concerns in terms of cloud storage and cloud computation is the next frequent topic of interest amidst the Top 35 works on cloud computing. This matter is considered in publications by Kaufman (2009), Wang et al. (2009), Li et al. (2010), Takabi et al. (2010), Wang et al. (2010), Yu et al. (2010), Grobauer et al. (2011), Subashini and Kavitha (2011), Wang et al. (2012), Wei et al. (2014), and Ali et al. (2015). Security issues were also highlighted by Dinh et al. (2013) in mobile cloud computing and Xu (2012) in cloud manufacturing.

Some of the other most cited papers, all categorized into the area of Computer Science, present a vision of using cloud computing in business (Buyya et al. 2009; Buyya et al. 2008) and in education (Sultan 2010). An in-depth conceptualiation of cloud computing was, in turn, provided by Foster et al. (2008), Zhang et al. (2010), and Wang et al. (2010).

Topic profiling was the next step in mapping the research field related to the issues of cloud computing. Its aim is to search for leading themes and point out the thematic boundaries of the field. According to the procedure, leading topics were studied through the prism of the most prolific authors, leading source titles and subject areas as well as core references.

## 5. Topic Profiling: Searching for Leading Themes

We searched for most often used keywords to identify the leading topics studied by the publications included in the research sample. The following expressions are listed among the Top 10 keywords: "cloud computing" (cited 11,762 times), "distributed computer systems" (2371), "computer systems" (1415), "cloud computing environments" (1234), "digital storage" (1007), "mobile cloud computing" (1005), "quality of service" (812), "information technology" (772), "algorithms" (763), and "scheduling" (753). To thoroughly study thematic boundaries of the research field related to cloud computing, leading topics manifested through most often cited keywords were analyzed through the prism of the most prolific authors, most productive journals, leading subject areas and core references.

### 5.1. Author—Topic Profiling

Analysis of keywords commonly used by the most productive researchers on cloud computing gives a valuable clue to determine which topics are considered as pivotal in this scientific area. Table 8 provides a brief overview of this dependence.

**Table 8.** Top most often cited keywords in research on cloud computing by an author.

| Author | Keywords |
| --- | --- |
| Buyya, R. (80) | cloud computing (59), quality of service (16), resource allocation (15), computer systems (12), data centers (11), distributed computer systems (11), service level agreements (10), cloud computing environments (9), computing environments (9), mobile cloud computing (9), taxonomies (9) |
| Huh, E.N. (48) | cloud computing (39), quality of service (11), information management (8), mobile cloud computing (8), resource allocation (8), task scheduling (7), thin clients (7), computer systems (6), data distribution (6), mobile devices (6), parallel computing (6), task scheduling (6) |
| Gani, A. (41) | cloud computing (26), mobile cloud computing (25), mobile devices (19), mobile applications (15), mobile computing (13), distributed computer systems (11), application offloading (7), distributed applications (7), distributed systems (6), taxonomies (6) |
| Li, J. (34) | cloud computing (25), cryptography (20), data privacy (12), digital storage (10), security of data (8), access control (7), attribute-based encryptions (7), outsourcing (7), attribute-based encryption (6) |
| Vasilakos, A.V. (29) | cloud computing (22), cryptography (5), surveys (5), big data (4), digital storage (4), distributed computer systems (4), mobile cloud computing (4), virtualizations (4), web services (4) |

Own study based on data retrieved from Scopus database (accessed 20 July 2018).

What stands out in Table 8 is the difference between keywords used by researchers depending on a main topic of their special interest. R. Buyya is active in a broad area of information and communication technologies with a major focus on cloud computing and grid computing on grounds of them being the most promising computer paradigms (Buyya et al. 2008, p. 5). The role of cloud computing also is highlighted by E.N. Huh. He devotes a lot of his attention to cloud computing as a provider of new possibilities for exploiting Internet of Things (Aazam and Huh 2014, p. 466), giving a birth to Cloud of Things (Aazam et al. 2014, p. 415). A. Gani's major area of interest is mobile cloud computing paradigm, notably in terms of application optimization strategies. The considerations, detailed in some of his most influential works, concentrate on challenges of heterogeneous nature of mobile cloud computing (Sanaei et al. 2014; Qi and Gani 2012). J. Li, similar to A.V. Vasilakos, narrowed down his research mostly to security concerns on cloud storage and cloud computation, the significance of which is demonstrated in the work by Ali et al. (2015) titled "Security in cloud computing: Opportunities and challenges". Unlike A.V. Vasilakos, J. Li, however, has a more pragmatic approach. His articles, to address security concerns, propose alternative constructions regarding, among other things, deduplication (Li et al. 2015), identity revocation (Li et al. 2015), and searchable encryption (Li et al. 2010).

### 5.2. Journal—Topic Profiling

To deepen the conducted analysis, the identification of the most often cited keywords used by the most prominent journal titles within the field was done. Table 9 presents the results of this analysis.

**Table 9.** Top most often cited keywords in research on cloud computing by a journal.

| Journal | Keywords |
|---|---|
| Lecture Notes in Computer Science (461) | cloud computing (386), distributed computer systems (74), computer systems (60), internet (45), quality of service (44),cloud computing environments (40), algorithms (36), computer science (33), cryptography (33), network security (32) |
| Applied Mechanics and Materials (296) | cloud computing (283), information technology (91), manufacture (56), computer systems (43), cloud computing environments (40), cloud computing technologies (34), information management (30), distributed computer systems (29), algorithms (24), digital storage (24) |
| Communications in Computer and Information Science (243) | cloud computing (222), computer systems (70), information technology (36), distributed computer systems (32), network security (25), computing environment (23), virtualizations (22), grid computing (21), cloud services (19), virtualization (19) |
| Advances in Intelligent Systems and Computing (238) | cloud computing (222), distributed computer systems (54), cloud computing environments (44), network function virtualization (35), digital storage (30), intelligent computing (26), artificial intelligence (25), quality of service (25), scheduling (25), big data (24) |
| Advanced Materials Research (187) | cloud computing (179), information technology (49), material science (33), computer systems (31), cloud computing technologies (25), cloud computing environments (16), cloud computing platforms (16), manufacture (16), information services (15), cloud services (13) |

Own study based on data retrieved from Scopus database (accessed 20 July 2018).

All the most prominent journal titles in the field share the expression "cloud computing" among top most often cited keywords. *Lecture Notes in Computer Science* is the unquestioned leader in regard to the frequency of occurrence of "cloud computing" (386) in its publications. This keyword rarely appears in such journals as *Applied Mechanics and Materials* (283), *Communications in Computer and Information Science* (222), *Advances in Intelligent Systems and Computing* (222), and even less frequently in *Advanced Materials Research* (179). In all but one Top 5 journals, "computer systems" and "distributed computer systems" are among the leading keywords. The keyword "computer systems" most frequently appears in two of the above-mentioned journals i.e., *Communications in Computer and Information Science* (70) and *Lecture Notes in Computer Science* (60). Similarly, the phrase "distributed computer systems" is

most frequent in the case of *Lecture Notes in Computer Science* (74). The fourth most often cited keyword appearing in works of *Lecture Notes in Computer Science*, *Applied Mechanics and Materials*, *Advances in Intelligent Systems and Computing* and *Advanced Materials Research* is "cloud computing environments" (140 times in total; 40, 40, 44 and 16, respectively). Furthermore, the keyword "information technology" was cited 176 times in total in three most prominent journal titles in the field: *Applied Mechanics and Materials* (91), *Advanced Materials Research* (49) and *Communications in Computer and Information Science* (36). *Applied Mechanics and Materials* and *Advanced Materials Research* focus more than the three remaining most prominent journals in the field on "cloud computing technologies" (keyword cited 59 times in total) and the issues related to the keyword "manufacture" (cited 72 times in total). On the other hand, e.g., in *Lecture Notes in Computer Science* and *Advances in Intelligent Systems and Computing*, the expression "quality of service" appears 69 times in total. Finally, the keywords: "algorithms" and "cloud services" are cited by two out of five most productive journals dealing with cloud computing. The phrase "algorithms" is cited 60 times in total by *Lecture Notes in Computer Science* and *Applied Mechanics and Materials*. The keyword "cloud services" appears 32 times in total in two journals: *Communications in Computer and Information Science* as well as *Advanced Materials Research*. The analysis of most often cited keywords in research on cloud computing by a journal, as presented in Table 9, allows stating that the most prominent source titles seem to have a bit different focus. For instance, *Advances in Intelligent Systems and Computing*, apart from highlighting a general cloud computing idea, seems to focus on the issues of network function virtualization, digital storage, artificial intelligence and scheduling as well as big data.

## *5.3. Subject Area—Topic Profiling*

Table 10 shows the most often cited keywords in the analyzed literature categorized by a subject area.

**Table 10.** Top most often cited keywords in research on cloud computing by a subject area.

| Subject Area | Keywords |
| --- | --- |
| Computer Science (10,627) | cloud computing (9046), distributed computer systems (1944), computer systems (1120), cloud computing environments (972), mobile cloud computing (867), digital storage (824), quality of service (703), cryptography (644), scheduling (614), resource allocation (602) |
| Engineering (4488) | cloud computing (3772), distributed computer systems (837), cloud computing environments (441), computer systems (385), mobile cloud computing (370), digital storage (363), information technology (293), scheduling (250), algorithms (249), network security (239) |
| Mathematics (2173) | cloud computing (1796), distributed computer systems (364), computer systems (240), cloud computing environments (218), scheduling (159), quality of service (153), digital storage (147), mobile cloud computing (139), algorithms (137), resource allocation (130) |
| Social Sciences (884) | cloud computing (707), distributed computer systems (120), education (58), computer systems (56), information systems (55), engineering education (54), digital storage (50), web services (48), e-learning (47), security of data (47) |
| Business, Management and Accounting (665) | cloud computing (521), distributed computer systems (62), information technology (61), information management (45), electronic commerce (42), computer systems (40), innovation (32), information systems (31), internet (30), cloud computing environments (29) |
| Decision Sciences (655) | cloud computing (573), distributed computer systems (131), cloud computing environments (85), big data (77), information management (55), digital storage (50), information systems (50), network function virtualization (50), mobile cloud computing (48), data handling (45) |

Own study based on data retrieved from Scopus database (accessed 20 July 2018).

All subject areas put both "cloud computing" and "distributed computer systems" as the most frequently appearing keywords. However, in terms of frequency, "cloud computing" is undoubtedly leading. It is most often cited in the subject area of Computer Science (9046 times), followed by Engineering (3772), Mathematics (1796), Social Sciences (707 times), Decision Sciences (573 times) and Business, Management and Accounting (521 times). The expression "distributed computer systems" is cited 1944 times in the area of Computer Science. Almost twice as rarely this keyword appears in the area of Engineering (837 times), and even less frequently in Mathematics (364 times), Decision Science (131 times), Social Sciences (120 times) and Business, Management and Accounting (62 times). The next most often cited keywords are "computer systems" and "cloud computing environments". All but one of the six subject areas presented in Table 10 are focused on these issues. Five of six subject areas highlight "computer systems" except for Decision Sciences. On the other hand, "cloud computing environments" is cited in all distinguished subject areas aside from Social Sciences, which is a bit more cantered on "engineering education" (54), "digital storage" (50), "e-learning" (47) and "security of data" (47). "Cloud computing environments" is most frequently cited in the publications assigned to Computer Science area (972). Again, less than half as often this keyword appears in the area of Engineering (441 times) and Mathematics (218 times). Much less frequently the keyword "cloud computing environments" is cited in the works published in subject areas such as Decision Science (85 times) and Business, Management and Accounting (29 times). "Digital storage" occurs in five of six subjects, excluding Business, Management and Accounting. Furthermore, "mobile cloud computing" occurs in four of six subject areas. This keyword is cited in all distinguished subject areas except for Social Sciences and Business, Management and Accounting. As regards "scheduling" and "information systems", these keywords appear in three of six subject areas. "Scheduling" is most often cited in the subject areas Computer Science (614 times), Engineering (250 times) and Mathematics (159 times). On the contrary, the keyword "information systems" occurs most frequently in the areas of Social Sciences (55 times), Decision Sciences (50 times) and Business, Management and Accounting (31 times).

Finally, there are few keywords that appear in two of six distinguished subject areas: "quality of service" (cited in total 856 times in the areas of Computer Science and Mathematics), "resource allocation" (cited in total 732 times in the fields of Computer Science and Mathematics), "information technology" (cited in total 354 times in the areas of Engineering and Business, Management and Accounting) and "information management" (cited in total 100 times in the area of Business, Management and Accounting and Decision Sciences).

*5.4. Core References—Topic Profiling*

Analysis of the Top 10 most cited publications indicates five core topics in the research field related to cloud computing: "cloud terminology" (Term), "cloud characteristics" (Char), "cloud computing architecture" (Arch), "advantages and disadvantages of cloud computing" (A&D), and "obstacles and opportunities of cloud computing" (O&O). Table 11 presents the complete results of the analysis for each of core publications.

As the above table shows, most of the core references focus on more than one topic. Among these the most highlighted topic is cloud computing architecture. Buyya et al. (2008), Buyya et al. (2009), and Beloglazov et al. (2012) concentrated on optimizing cloud architecture with regard to resource allocation management. Other examples show the usefulness of utilizing such architectural frameworks as CloudSim (Calheiros et al. 2011) and Eucalyptus (Nurmi et al. 2009). Particular attention is also given to basic conceptualization of cloud computing as it still is treated as a new paradigm. For instance, Buyya et al. (2009) indicated key characteristics distinguishing cloud computing from the closest technologies to it, i.e. grid computing and cluster computing. Foster et al. (2008) carried out a more thorough analysis by comparing cloud computing to grid computing in terms of a business model, architecture, resource management, programming model, application model, and security model. Marston et al. (2011), in turn, establish terminology of cloud computing as a guide to outlining

strengths, weaknesses, opportunities and threats of deploying cloud. A similar solution aws applied by Armbrust et al. (2010) who specified obstacles and opportunities of leveraging cloud computing. Subashini and Kavitha (2011) narrowed this discussion down by describing security issues in terms of cloud service delivery models.

**Table 11.** Top most often cited core references in research on cloud computing by a topic.

| Reference | Citations | Topic | | | | |
|---|---|---|---|---|---|---|
| | | Term | Char | Arch | A&D | O&O |
| Armbrust et al. (2010) | 4512 | | x | | | x |
| Buyya et al. (2009) | 2963 | | | x | | |
| Calheiros et al. (2011) | 1816 | x | x | x | | |
| Foster et al. (2008) | 1627 | x | | | | x |
| Zhang et al. (2010) | 1383 | x | x | x | | |
| Subashini and Kavitha (2011) | 1148 | x | x | x | x | |
| Beloglazov et al. (2012) | 1109 | | | x | | |
| Buyya et al. (2008) | 1078 | | x | x | | |
| Nurmi et al. (2009) | 1050 | | | x | | |
| Marston et al. (2011) | 937 | x | x | | x | x |

Own study based on data retrieved from Scopus database (accessed 20 July 2018).

Summing up, the analysis of topic profiling presented above proved that cloud computing is a complex and multi-faced concept. However, there are some areas that gain attention of scholars from different fields. The main topics of interest that have been identified by topic profiling, through the prism of the most prolific authors, most productive journals, leading subject areas and core references, include: (1) conceptualization of cloud computing as a new paradigm, with particular attention given to the issues of its application possibilities and challenges related to its heterogeneous nature; (2) the concept of distributed computer systems aiming at maximizing performance by connecting users and IT resources in a cost-effective and reliable manner; (3) security concerns regarding cloud computing; (4) cloud computing environments; and (5) quality of cloud computing service.

## 6. Conclusions

Both the number of publications and the number of citations they have received prove the great interest of researchers in studying various issues related to cloud computing. Through general publication profiling, this study attempted to answer the first of posted research questions: What are the main contributors to the scholarly community interested in cloud computing, including countries, research institutions, journals and authors? The findings indicate that, among the most productive countries, the main contributors—leaders regarding both quantity and quality of publications—are China, India and the United States. As far as institution profiling analysis is concerned, the most productive research institutions represent the countries identified as the main contributors to the research field. In regard to quantitative perspective, quite a strong fragmentation of research output is observed. Similarly, as regards the visibility of publications one can notice a diversity of the research output. In terms of visibility, the most productive universities are: the University of Melbourne, the University of Malaya, Arizona State University, Tsinghua University and Chinese Academy of Sciences. These findings are quite consistent with the findings of Heilig and Voß (2014) obtained five years ago. The aforementioned authors as the main contributors identified China and the United States and among most productive institutions, they pointed out University of Melbourne, Beijing University of Posts and Telecommunications, Tsinghua University or Arizona State University. However, the study presented in this paper confirms the progress of cloud computing research in India, which has become a fast growing market in terms of both cloud computing research and industry.

The most productive source titles within the field are: *Lecture Notes in Computer Science* (the highest number of publications) and *Communications in Computer and Information Science*. Nevertheless, taking

the publication visibility perspective, *Future Generation Computer Systems* (the highest number of received citations) and *Procedia Computer Science* also need to be mentioned. In regard to the authors contributing to the field, R. Buyya from the University of Melbourne is the most prolific author dealing with the issues of cloud computing. Again, this result is the same as provided by Heilig and Voß in 2014, although certainly the number of R. Buyya's publication has increased since 2014.

Secondly, the paper recognizes and explores the leading subject areas in the field. The analysis of subject areas allows responding to the second research question: How much multidisciplinary is research in the field and how is the research output distributed across subject areas? The research output on cloud computing is distributed among 25 subject areas, while those of the highest number of publications are in: Computer Science, Engineering, Mathematics, Social Sciences, Business, Management and Accounting and Decision Sciences. As far as the main subject areas in the field of cloud computing research are concerned, nothing has changed since 2014. The same areas dominate, having the same order as provided by Heilig and Voß five years ago. Subject area profiling has included the identification of the most prominent source titles, top contributing authors, and core references in each of the six leading subject areas. The analysis of the most productive journals associated with the aforementioned key subject areas shows that in each subject area there is a group of prominent source titles dedicated to it. Only a few of them are associated with more than one subject area and no more than two of them. Leaving aside the division of the most productive journals among the subject areas, the unquestioned leader in terms of the quantity of publications related to cloud computing is *Lecture Notes in Computer Science*. This definitely has changed since 2014. Through their scientometric analysis, Heilig and Voß in 2014 found *Future Generations Computer Systems* as the journal leading in terms of number of cloud computing publication per year (Heilig and Voß 2014). Going further, R. Buyya is the most prolific researcher in the subject areas of Computer Science and Mathematics. The leading position of R. Buyya has not changed since the research conducted by Heilig and Voß. In the first of the above-mentioned subject categories, the main authors are: E.N. Huh, J. Li, A. Gani, A.V. Vasilakos, M. Qiu, and X. Chen. In the area of Engineering, the highest number of publications was produced by S. Yamada, while, in the area of Mathematics, the most prolific authors are R. Buyya and A. Gani, A. In the case of Social Sciences, the most productive authors are M. Yuvaraj and N. Sultan, while, in Business, Management and Accounting, the most prolific researcher is V. Ratten. In the area of Decision Sciences, the highest number of publications i produced by E.N. Huh. Two references related to cloud computing that received main attention (are most highly cited) are: the work of Armbrust et al. (2010), representing Computer Science subject area, which gathered 4512 citations and the paper by Buyya et al. (2009), also representing Computer Science subject area, which received 2963 citations. In the catalogue including the publications in six subject areas under the study cited over 500 times (in total 15 titles), there is only one publication shared by three subject areas and three papers shared by two subject areas.

Thirdly, the conducted research enabled answering the third of research questions formulated in the paper: What are the leading topics of studies in the field? The study analysed the most often cited keywords to point out the predominant themes related to cloud computing. Among these most often used keywords, there are such expressions as: "cloud computing", "distributed computer systems", "computer systems", "cloud computing environments", "digital storage", "mobile cloud computing", "quality of service", "information technology", and "algorithms" and "scheduling". Topic profiling enabled finding leading themes for the most prolific authors, main source titles and subject areas as well as core references. The publications by R. Buyya from the University of Melbourne, considered to be the most prolific author, are at the top of publications related to 6 out of 10 leading topics within the field (i.e., "cloud computing", "quality of service", "computer systems", "distributed computer systems", "cloud computing environments" and "mobile cloud computing").

The most prominent journals in the field share the keyword "cloud computing". Nevertheless, *Lecture Notes in Computer Science* is the unquestioned leader in terms of the frequency of occurrence of "cloud computing" (386) in its publications. Moreover, all but one of the five top journals show

a lot of interest in the issues related to "computer systems" and "distributed computer systems" as well as "cloud computing environments". Finally, "information technology" occurs in three of the most prominent source titles in the field. As regards the perspective of the leading subject areas, "cloud computing" and "distributed computer systems" are the most cited keywords in all main subject areas. However, considering the number of citations in all subject areas, "cloud computing" is definitely the leading keyword. Moreover, the keywords "computer systems" and "cloud computing environments" are found in five of six subject areas. Both "computer systems" and "cloud computing environments" are most frequently cited in the publications assigned to Computer Science area. Analyzing core references corresponding to top keywords within the field allowed identifying five leading research topics on cloud computing, namely conceptualization of cloud computing as a new paradigm, the concept of distributed computer systems, security concerns regarding cloud computing, cloud computing environments and cloud computing service quality. The findings referring to the main research topics in the field of cloud computing research differ slightly from those of Heilig and Voß. In particular, cloud computing environments seem to be a new research area. In addition, distributed computer systems as a research topic has gained in importance since 2014.

Concluding, the conducted study makes the following contributions. First, it provides a holistic view on cloud computing research over the last decade. Secondly, by providing a holistic view on a research field from a meta-perspective, it facilitates the development of the field through better understanding of the multi-faced concept of cloud computing and identifying the areas and some suggestions for future research efforts. Nevertheless, we are aware of the limitations of our study. Firstly, research profiling is the only method applied in this study to map the field. Consequently, further studies with the use of other bibliometric methods and techniques are recommended to increase the objectivity of the findings through research methodology triangulation. Moreover, these quantitative literature studies may be combined with qualitative surveys, which generally ensure a more thorough analysis of selected aspects. Secondly, we decided to take an overall picture of the whole research field. Nevertheless, due to its size (more than 14,000 publications included into the research sample) and diversity (manifested thorough the number of subject areas), some superficiality of such a study is a natural consequence. To mitigate this weakness, and ensure a more thorough exploration, more detailed profiling studies in particular subject areas are recommended. Thirdly, limiting the sampling process to only publications with the expression "cloud computing" in their titles might have excluded some valuable papers referring to the issues of cloud computing in published in journals important for the research field (e.g., *IEEE Transactions on Cloud Computing* or *Journal of Cloud Computing*) or presented at relevant conferences (International Symposium in Cluster, Cloud and Grid Computing CCGrid, International Conference on Cloud Computing CLOUD, and International Conference on Cloud Computing and Services Science CLOSER). Consequently, the idea of focusing the attention on publications relating directly to the issues of cloud computing (what is manifested in article titles) could have narrowed the research sample too much. Therefore, it is worth considering the replication of the study based on the enlarged sample consisting of publications including the expression "cloud computing" in their titles, keywords and abstracts. As of 20 March 2019, in Scopus database, there are 64,071 such documents. To deal with such a numerous research sample, clustering techniques supported with IT applications (e.g., CitNetExplorer and VOSviewer) could be employed to conduct topic profiling (cf. Van Eck and Waltman 2017). Finally, using only one database (i.e., Scopus) for the sampling process should be considered as another limitation, in particular in regard to neglecting the research output published in languages other than English, as Scopus is very much biased towards indexing English language literature. That is why we recommend some additional literature reviews focusing on exploring non-English research publications in the field. Moreover, the presented study inspires conducting further research regarding identification of connections and differences between the economic development of the dominant countries and their research as well as scientific development.

**Author Contributions:** A.S. initiated the project and was its leader. A.L. conceptualized the paper, designed methodology as well as retrieved and prepared data for analysis. R.B. outlined theoretical background. All the authors contributed to data analysis and interpretation as well as writing and editing the manuscript.

**Funding:** This research received no external funding.

**Acknowledgments:** The authors are grateful to the anonymous reviewer for pointing out this limitation. The authors are grateful to the anonymous reviewer for this suggestion.

**Conflicts of Interest:** The authors declare no conflict of interest.

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
