# Peer review of "Cloud Computing Research Profiling: Mapping Scholarly Community and Identifying Thematic Boundaries of the Field"

_socsci, doi:10.3390/socsci8040112_

Round 1

Reviewer 1 Report

The paper presents an interesting study from different points of scientometric views of a set of articles in cloud computing. 

Several issues can be mentioned related to the approach:

(1) the set of papers have been selected searching 'cloud computing' in the title; by this decision a large and significant set of papers that are important for field have been ignored and it is expected that the ones that are applying cloud computing in different fields have another weight (a proof is the presence in the analysis of the journals related to advanced materials or applied mechanics). The set should have included the journals and conferences that are relevant for cloud computing (like IEEE Transactions on Cloud Computing, Springer Journal of Cloud Computing and conferences like CCGrid, CLOUD, CLOSER etc to which almost all papers are about cloud computing but with subtopics of it). A most relevant selection should have been done based on the inclusion as keywords of cloud computing. Deviations like hybrid cloud, cloud federation are also significant.  

(2) the paper should have start from the previous analysis with the same topic like Leonard Heilig and Stefan Voss , "A Scientometric Analysis of Cloud Computing Literature", IEEE Transactions on Cloud Computing vol 2, issue 3, 266-278, DOI 10.1109/TCC.2014.2321168, and identify the differences (e.g,. in the last 4 years)

Other issues:

- page 4, line 118 and 122, leadership in terms of quality is not necessarily related to the citations (can be negative too), rather visibility of the work. 

- section 5 should have been concluded with a list of topics of interest

- last two references from "Advanced Materials Research" are not relevant

Author Response

Response to Reviewer 1 comments

The paper presents an interesting study from different points of scientometric views of a set of articles in cloud computing. 

Several issues can be mentioned related to the approach:

Point 1: the set of papers have been selected searching 'cloud computing' in the title; by this decision a large and significant set of papers that are important for field have been ignored and it is expected that the ones that are applying cloud computing in different fields have another weight (a proof is the presence in the analysis of the journals related to advanced materials or applied mechanics). The set should have included the journals and conferences that are relevant for cloud computing (like IEEE Transactions on Cloud Computing, Springer Journal of Cloud Computing and conferences like CCGrid, CLOUD, CLOSER etc to which almost all papers are about cloud computing but with subtopics of it). A most relevant selection should have been done based on the inclusion as keywords of cloud computing. Deviations like hybrid cloud, cloud federation are also significant.  

Response 1: The research sampling process was purposely limited to the title search in order to concentrate the attention on publications relating directly to the issue of cloud computing. For the same reason, we also excluded any other expressions linked with the topic e.g. ‘hybrid cloud’, ‘cloud federation’. Nevertheless, we are aware of limitations resulting from such a design of the research sampling process, which we made more explained in the conclusion section.

We are grateful to the reviewer for pointing out negative consequences of research sampling limitations. However, we decided to maintain the existing sampling in this paper, and extend it in future research, employing bibliometric methods relevant for analyzing big size samples (e.g. clustering technique).

Point 2: the paper should have start from the previous analysis with the same topic like Leonard Heilig and Stefan Voss , "A Scientometric Analysis of Cloud Computing Literature", IEEE Transactions on Cloud Computing vol 2, issue 3, 266-278, DOI 10.1109/TCC.2014.2321168, and identify the differences (e.g,. in the last 4 years)

Response 2: We are grateful to the reviewer for pointing out the work of Leonard Heilig and Stefan Voss (2014). We studied this paper and added in the work the statements comparing the findings of Leonard Heilig and Stefan Voss in regard to main contributors to the field, main subject areas as well as leading research topics in the field in order to illustrate the changes taking place in the field of cloud computing research.

Other issues:

Point 3: - page 4, line 118 and 122, leadership in terms of quality is not necessarily related to the citations (can be negative too), rather visibility of the work. 

Response 3: We complied with the suggestion and changed wording related to the visibility of analyzed publications.

Point 4: - section 5 should have been concluded with a list of topics of interest

Response 4: We complied with the suggestion and at the end of section 5 we added a paragraph pointing out the main topics of interest that have been identified by topic profiling, through the prism of the most prolific authors, most productive journals, leading subject areas and core references.

Point 5: - last two references from "Advanced Materials Research" are not relevant

Response 5: We complied with the suggestion and removed these items from the manuscript.

Reviewer 2 Report

The manuscript develops an interesting and valid topic. The authors know the subject and the most important bibliography published. The paper presents a very wide perspective on profiling research in the Cloud Computing area. The authors present a lot of data in this area, based on a large research sample. However, the manuscript has some important weaknesses. The abstract includes information about Scopus database, while the authors mention the WoS database also in content of paper. This should be clarified.

In the body of the article under Figure 1, the authors describe the results and refer to figure 3. Was it intentional? Next, the authors present a polynomial function for the efficiency of research and citations. However, the function requires better interpretation. The authors write about the division of the rest of the countries into three categories. What categories? Why do they cover only selected countries? In what group are the rest countries for N = 127?

The authors include research questions in the introduction. However, it is difficult to point to unambiguous answers for them in the summary. A better combination of questions and conclusions is recommended.

The results show the domination of Asian countries. How can you explain this phenomenon? What are the conclusions and recommendations based on the domination of Asian universities? What makes them leaders in research? The conclusions do not include a answer to the questions: How the analysis contributes to the development of the scientific community? How can the results support scientists and future research?

At the same time, it is recommend a further research in the area of identification of connections and differences between the economic development of the dominant countries and their research and scientific development.

Author Response

Response to Reviewer 2 comments:

Point 1: The manuscript develops an interesting and valid topic. The authors know the subject and the most important bibliography published. The paper presents a very wide perspective on profiling research in the Cloud Computing area. The authors present a lot of data in this area, based on a large research sample. However, the manuscript has some important weaknesses. The abstract includes information about Scopus database, while the authors mention the WoS database also in content of paper. This should be clarified.

Response 1: We provided necessary clarification in the abstract.

Point 2: In the body of the article under Figure 1, the authors describe the results and refer to figure 3. Was it intentional? Next, the authors present a polynomial function for the efficiency of research and citations. However, the function requires better interpretation. The authors write about the division of the rest of the countries into three categories. What categories? Why do they cover only selected countries? In what group are the rest countries for N = 127?

Response 2: Ambiguous reference to the figure was an omission, we corrected it.

We removed phrases about polynomial functions, while maintaining qualitative description of publication and citations trends.

In regard to the comment referring the description of top most productive countries in research on cloud computing, we concentrate on the leaders – top 10 most productive countries (the statement on the page 3). That is why we analyze just 10 countries included in table 1, grouped into three categories described on page 4. We do not analyze the remaining 117 countries identified in the research.

Point 3: The authors include research questions in the introduction. However, it is difficult to point to unambiguous answers for them in the summary. A better combination of questions and conclusions is recommended.

Response 3: We complied with the suggestion. In “Conclusions” section we provided the explicit answers related to all three research questions formulated in the “Introduction” section.

Point 4: The results show the domination of Asian countries. How can you explain this phenomenon? What are the conclusions and recommendations based on the domination of Asian universities? What makes them leaders in research? The conclusions do not include a answer to the questions: How the analysis contributes to the development of the scientific community? How can the results support scientists and future research?

Response 4: We complied with both aforementioned suggestions. The paragraph related to explaining the dominant position of Asian countries and Universities in the field of cloud computing research was added to the section 3.2. (page 5-6).

In “Conclusions” we also made an explicit statements concerning the contributions of the presented study as well as further research areas (page 19).

Point 5: At the same time, it is recommend a further research in the area of identification of connections and differences between the economic development of the dominant countries and their research and scientific development.

Response 5: The authors are grateful to the anonymous reviewer for pointing out this recommendation. Such a direction of further research in the field has been mentioned in the “Conclusions” section.

Reviewer 3 Report

The paper provides a statistical analysis of the publications on cloud computing.

Data are analyzed in terms of number of papers, citations, authors, institutions, subject areas.

Author Response

Response to Reviewer 3 comments:

Point 1: The paper provides a statistical analysis of the publications on cloud computing.

Data are  analyzed in terms of number of papers, citations, authors, institutions, subject areas.

Response 1: The authors are grateful to the anonymous reviewer for the effort related to reviewing the paper.

Round 2

Reviewer 1 Report

The paper content was improved following the reviewer comments. 

No further concerns are generated by the new text.